# Investigating the Antimicrobial Potential of 560 Compounds from the Pandemic Response Box and COVID Box against Resistant Gram-Negative Bacteria

**DOI:** 10.3390/antibiotics13080723

**Published:** 2024-08-01

**Authors:** Rita de Cássia Cerqueira Melo, Aline Andrade Martins, Andressa Leite Ferraz Melo, Jean Carlos Pael Vicente, Mariana Carvalho Sturaro, Julia Pimentel Arantes, Luana Rossato, Gleyce Hellen de Almeida de Souza, Simone Simionatto

**Affiliations:** Health Sciences Research Laboratory, Federal University of Grande Dourados (UFGD), Dourados 79804970, Mato Grosso do Sul, Brazil; riitaccmelo@gmail.com (R.d.C.C.M.);

**Keywords:** antimicrobial resistance, Gram-negative bacteria, drug screening, multidrug-resistant resistance

## Abstract

Antimicrobial resistance (AMR) has emerged as a significant threat to public health, particularly in infections caused by critically important Gram-negative bacteria. The development of novel antibiotics has its limitations, and therefore it is crucial to explore alternative strategies to effectively combat infections with resistant pathogens. In this context, the present study investigated the antibacterial potency of 560 compounds against the multidrug-resistant (MDR) strains of *Klebsiella pneumoniae* and *Serratia marcescens*. The evaluated compounds were selected from the Pandemic Response Box (PRB) and COVID Box (CB) and subjected to assays to determine the inhibitory concentration (IC), minimum bactericidal concentration (MBC), and biofilm formation. Further, the effects of these compounds on membrane integrity were assessed through protein quantification. Several of the evaluated compounds, including fusidic acid, MMV1580853, and MMV1634399, exhibited a significant reduction in biofilm formation and growth in *K. pneumoniae*. Trimethoprim exhibited potential against *S. marcescens*. The IC values of the compounds indicated significant microbial growth inhibition at various concentrations. These findings underscore the potency of the existing antibiotics and novel compounds in combating the MDR strains of bacteria. The importance of reconsidering the known antibiotics and utilizing drug repositioning strategies to address the increasing risk of AMR is highlighted.

## 1. Introduction

Antimicrobial resistance (AMR) in critical pathogens has emerged as a major public health concern globally, with significant implications in terms of morbidity, mortality, and healthcare costs [1]. Considering the urgency of addressing AMR, the World Health Organization (WHO) has identified a list of priority pathogens, including critical multidrug-resistant (MDR) bacteria, such as *Klebsiella pneumoniae* and *Serratia marcescens* [2,3]. The WHO recommends prioritizing the research and development efforts to identify novel treatment options for infections caused by these pathogens that greatly threaten human health owing to their high levels of resistance and the severity of the infections they cause [4]. The multidrug-resistant (MDR) strains of *K. pneumoniae* and *S. marcescens* are clinically and epidemiologically significant as they cause severe and often life-threatening infections, especially in healthcare settings [5]. The significant impact of these pathogens on patient outcomes and the challenges they pose to infection control in healthcare settings highlight the importance of focusing on MDR strains [6,7].

The development of effective treatments against these MDR bacteria is essential to improve clinical outcomes and manage the spread of infections with resistant bacteria. In this context, adhering to the WHO recommendations could facilitate the effective coordination of efforts worldwide to combat the increasing threat of AMR. However, the development rate of novel antibiotics has decelerated in recent times, with only two novel classes of antibiotics reaching the market in the last years [8]. This stagnation underscores the urgent requirement for developing alternative strategies to combat AMR. One promising approach is to repurpose existing drugs originally developed for other bacterial strains to treat infections with MDR bacteria. This strategy could accelerate the availability of novel treatment options against MDR bacteria by leveraging the pre-existing clinical data and streamlining the preclinical and clinical testing phases. Institutions providing open-access drug libraries, such as the Pandemic Response Box (PRB) and the COVID Box (CB), play a crucial role in facilitating the discovery of novel antimicrobial therapies through potential repositioning [9,10].

The present study involved evaluating a total of 560 compounds from the Medicines for Malaria Venture (MMV) Pandemic Response Box and COVID Box for their antibacterial potential against the multidrug-resistant (MDR) strains of *K. pneumoniae* and *S. marcescens*.

## 2. Results

### 2.1. Antimicrobial Susceptibility Evaluation

Antimicrobial susceptibility testing of strains showed resistance to multiple antibiotics (Table 1). The inhibitory effects of compounds from the MMV PRB and CB against critical MDR pathogens were evaluated. Most compounds exhibited limited inhibitory potential (Figure 1). For the carbapenem- and polymyxin-resistant *K. pneumoniae* (CPR-*Kp*) strain, thirty-eight compounds exhibited inhibition percentages above 80%, while three compounds exhibited inhibitory percentages above 90% (Figure 1A, including fusidic acid (91.28%), MMV1580853 (90%), and MMV1634399 (90%). Against the carbapenem-resistant *K. pneumoniae* (CR-*Kp*) strain, four CB compounds exhibited inhibition rates greater than 80% (Figure 1B), namely oxyclozanide (93.12%), doxorubicin (91.74%), and niclosamide (89.00%), while among PRB compounds, only gepotidacin (91.84%) and MMV1634402 (90.78%) exhibited inhibition percentages above 80% (Figure 1A). The IC values of gepotidacin and MMV1634402 against CR-*Kp* were 0.625 µM and 10 µM, respectively (Figure 2c). None of the compounds from CB exhibited an inhibition percentage over 80% against CR-*Kp* (Figure 1B).

For carbapenem-resistant *S. marcescens* (CR-*Sm*), trimethoprim exhibited an inhibition percentage of 82.14%, while sitafloxacin, MMV1580849, and eravacycline exhibited 79.60%, 79.37%, and 71.16% inhibition, respectively (Figure 1B). Trimethoprim exhibited an MIC value of 10 µM against CR-*Sm* (Figure 2d). None of the compounds from CB exhibited an inhibition percentage of over 80% against CR-*Sm* (Figure 1B). 

### 2.2. Survival Curves

Growth curves were obtained to evaluate the effectiveness of compounds that initially showed activity against the CPR-*Kp*, CR-*Kp*, and CR-*Sm* strains for 24 h. Among the PRB compounds, MMV1634399, MMV1580853, and fusidic acid exhibited efficacy against CPR-*Kp* after 10 h (Figure 3a). Gepotidacin exhibited efficacy initially, followed by a decline in the efficacy against CR-*Kp* (Figure 3c). Trimethoprim from PRB exhibited consistent efficacy in the inhibition of CR-*Sm* growth (Figure 3d). A summary of the compounds with the best activities, showing the percentage of inhibition, the IC, the disease set, and the chemical structure of the compound can be viewed in Table 2.

### 2.3. Anti-Biofilm Assay

MMV1634399, fusidic acid, and MMV1580853 were observed to significantly inhibit biofilm formation of CPR-*Kp* (Figure 4a). Doxycycline, tetracycline, doxorubicin, oxyclozanide, and niclosamide led to significantly lowered biofilm formation, indicating their inhibitory activities against CPR-*Kp* (Figure 4b). In contrast, gepotidacin and MMV1634402 exhibited low anti-biofilm efficacy for CR-*Kp*, with results similar to those observed for the positive control (Figure 4c). No significant inhibitory effect of the compounds on biofilm formation was observed for CR-*Sm* (Figure 4d).

### 2.4. Membrane Integrity Assay

The effects of the evaluated compounds on bacterial cell membrane integrity were determined using protein leakage quantification as an indicator. However, when the bacteria were treated with these compounds, no protein leakage was observed (Figure 5).

## 3. Discussion

The antibacterial potential of the compounds from PRB and CB against the MDR strains of *K. pneumoniae* and *S. marcescens* was evaluated. The discovery of novel compounds with the ability to control CRP-*Kp* growth was an important finding of the present study. Since this strain is resistant to carbapenems and polymyxins, it is considered a major challenge in the field of medicine in terms of the management and control of AMR [11,12]. Several other compounds were revealed to be promising in terms of their inhibitory activities, highlighting the potential of both under-investigated molecules and antibiotics to be applied against MDR bacteria. Among the 560 compounds evaluated in the present study, three were particularly relevant for CPR-*Kp*, with promising results obtained in the IC, MBC, and biofilm assays: fusidic acid, MMV1580853, and MMV1634399. It is noteworthy that the CPR-*Kp* strain is the most difficult to treat due to its resistance profile and is, therefore, on the priority list of the WHO [3]. To our knowledge, this is the first study to evaluate these box compounds against the MDR bacteria with intrinsic or chromosomal resistance to polymyxin.

Fusidic acid is a unique antibiotic derived from the fungus *Fusidium coccineum*, with a steroid-like structure but no corticosteroid effects [13,14]. It inhibits bacterial protein synthesis by binding to elongation factor G, preventing ribosomal translocation [15]. Fusidic acid is particularly effective against *Staphylococcus aureus*, including methicillin-resistant strains, and is used to treat various skin and soft tissue infections [16,17]. Its structure allows for high skin penetration, and it shows low resistance and no cross-resistance with other antibiotics [13]. Topical combinations with corticosteroids are useful in treating atopic dermatitis with suspected staphylococcal infection [16]. A new oral dosing regimen involving front-loading followed by maintenance doses has been developed to minimize resistance development in monotherapy [15]. The application of fusidic acid for the treatment of staphylococcal [13] and enterococci infections [15] was described. In addition, fusidic acid showed a synergistic effect when combined with polymyxin B against *K. pneumoniae* and *E. coli* [14]. Thus, these results corroborate its potential repurposing against Gram-negative MDR bacteria identified in our study. 

MMV1580853 is a bisamidine derivative that acts on undecaprenyl diphosphate synthase (UPPS), which is a critical enzyme for bacterial cell wall synthesis [18]. Although there are no ongoing clinical studies, preliminary findings suggest that MMV1580853 may have the potential to disrupt UPPS, which could in turn affect the production of essential components of the bacterial cell wall, ultimately leading to bacterial cell death [19,20]. Pre-clinical evaluations utilizing murine models of staphylococcal infection have indicated that MMV1580853 is generally safe, despite its potential toxicity. Furthermore, these evaluations have demonstrated the compound’s reproducibility in vivo [21]. MMV1580853 may also act as an active inhibitor of the main protease (Mpro) of SARS-CoV-2, which is crucial for the virus’s replication. Further studies in molecular dynamics and interactions between MMV1580853 and Mpro, with high free energy binding, suggest that it could be a promising therapeutic agent for treating the novel coronavirus disease. Additionally, in vivo tests with diamidine derivatives similar to MMV1580853 in *Trypanosoma brucei* infection models have shown promising results, with 100% survival in treated mice at 3 mg/kg intraperitoneally. These results suggest that MMV1580853 may have potential in antiviral and antiparasitic therapies. We tested a series of amidine and related compounds against *Trypanosoma brucei* [22].

MMV1634399, also known as HT61 (4-methyl-8-phenoxy-1-(2-phenylethyl)-2,3-dihydropyrrolo[3,2-c]quinoline), is a quinoline derivative that has been demonstrated to exhibit bactericidal activity against both methicillin-sensitive and methicillin-resistant *Staphylococcus aureus* [23]. It has been demonstrated that HT61 acts by increasing the permeability of bacterial membranes, resulting in the depolarization and release of intracellular constituents, which indicates significant structural damage to the bacterial membrane [24]. Moreover, HT61 has been tested in combination with other antimicrobials, including neomycin, gentamicin, mupirocin, and chlorhexidine, demonstrating significant synergistic effects against clinical strains of MSSA and MRSA, both in vitro and in vivo [24,25]. In murine models of lung infection, the combination of HT61 with tobramycin has been demonstrated to effectively deplete the bacterial burden, thereby highlighting its potential as an adjuvant agent to enhance the efficacy of conventional antibiotics [26]. Additionally, it has been shown to exhibit inhibitory activity against fungi [27].

The present study demonstrated that MMV1634399 is an effective agent for the reduction of bacterial growth and the inhibition of biofilm formation. This indicates that the compound targets essential bacterial processes that are independent of any impact on the cell membrane. The capacity of MMV1634399 to repress biofilm formation, which serves to protect bacteria against assaults from the immune system and antibiotics, underscores its potential as a valuable contender for future investigation. Drug repurposing, or reprofiling, is the process of identifying new therapeutic applications for existing pharmaceutical agents. While in vitro (test tube or cell culture) and pre-clinical (animal) studies can provide promising initial results, the process of translating these findings into clinical use is inherently rigorous and lengthy. This process comprises several phases of clinical trials, the objective of which is to guarantee the drug’s therapeutic efficacy and safety for human use [28,29,30]. The drug discovery and development are considered complex, time-consuming, and resource-intensive. They require multidisciplinary expertise and innovative approaches [31].

*S. marcescens* is intrinsically resistant to a broad spectrum of antibiotics, including carbapenems, polymyxins, penicillins, cephalosporins, macrolides, nitrofurantoin, and polymyxins. When bacteria develop resistance to carbapenems, treating infections becomes significantly more challenging. In the present study, trimethoprim was evaluated for its inhibitory effects on CR-*Sm* bacterial growth, and the results revealed its efficacy in this regard. Scottish guidelines for treating UTIs, especially those caused by multidrug-resistant (MDR) organisms, recommend the nitrofurantoin trimethoprim [32], which, despite the risks of resistance, continues to be effective against several strains, including CR-*Sm*. Trimethoprim is used commonly in combination with sulfamethoxazole and is considered valuable for urinary infections due to its high urinary concentration [33]. However, trimethoprim should be used only after susceptibility testing due to its potential adverse effects, although it could be reconsidered as an option when clinical choices are limited, due to its demonstrated efficacy against the resistant strains [34].

To identify possible mechanisms of action of the compounds tested, we conducted the protein extravasation test, which is often used as an indicator of the integrity of the cell membrane. This leakage means significant damage to the cell envelope, which includes the cell membrane and, in some cases, the cell wall. Minor damage may not lead to detectable protein leakage but can still impair cell function and viability. Furthermore, cells have mechanisms to quickly repair minor damage to the envelope, preventing protein leakage. However, the cell envelope can still allow leakage of smaller molecules, such as ions and metabolites, without proteins. Therefore, the absence of protein extravasation in our assays does not rule out the presence of damage [35,36].

Although our results were promising, this study had some limitations, including a limited number of strains tested and the assays only being performed in duplicate. This study was also constrained by the limited number of investigations involving these compounds with antibacterial activity. Further research is needed to elucidate their mechanisms of action. Additionally, in vivo assays need to be conducted to confirm that they are effective in an infection model.

The findings of the present study highlight the potential of both well-established and novel compounds for application in combating MDR bacterial strains. The efficacy demonstrated by fusidic acid, MMV1580853, and MMV1634399 against *K. pneumoniae* highlighted their potential as valuable candidates for further research, particularly to address the challenges raised by biofilm formation and antibiotic resistance. The notable performance of trimethoprim against *S. marcescens* suggested the requirement for continued exploration of its application potential in the treatment of resistant strains. The findings of the present comprehensive evaluation of some compounds emphasize the critical role of drug repositioning and exploration of synergistic drug combinations in developing effective treatments against MDR infections, offering hope for improved clinical outcomes in the face of escalating antibiotic resistance.

## 4. Materials and Methods

### 4.1. Bacterial Isolates

Previously characterized carbapenem-resistant *K. pneumoniae* (CR-*Kp*), carbapenem-polymyxin-resistant *K. pneumoniae* (CPR-*Kp*), and carbapenem-resistant *S. marcescens* (CR-*Sm*) strains were used in the present study. All strains used in this study were clinical isolates from patients admitted to a tertiary hospital in the Central-West region of Brazil, serving as a reference center for 32 cities [11,37]. Species identification was performed using a Phoenix^®^ Automated System (BD Diagnostic Systems, Sparks, MD, USA), followed by validation through matrix-assisted laser desorption/ionization time-of-flight (MALDI-TOF) mass spectrometry using a Microflex LT spectrometer (Bruker Daltonics, Billerica, MA, USA). Antimicrobial susceptibility profiles were determined next, including the determination of the minimum inhibitory concentrations (MICs), using the broth microdilution method by following the guidelines of the Clinical and Laboratory Standards Institute (CLSI) [38]. Whole-genome sequencing was then conducted to explore the molecular mechanisms underlying the resistance to carbapenems and polymyxin B.

### 4.2. Screening the Antibacterial Compounds

The Pandemic Response Box (PRB) was provided by Medicines for Malaria Venture (MMV, Geneva, Switzerland) and the Drugs for Neglected Diseases initiative (DNDi, Geneva, Switzerland), and the COVID Box by DNDi. The Pandemic Response Box is made up of 201 antibacterials, 153 antivirals, and 46 antifungals. The biological activity of the compounds was not confirmed by the MMV/DNDi partners, and the selection was made based on information available in the literature [39]. The compounds present in the 10 mM stock solution plate to a concentration of 1 mM in 100% DMSO were diluted in 90 µL of DMSO to prepare the stock solution and then diluted in the working solution in intermediate plates with a concentration range of 20–0.039 µM. The screening assay was performed by described for [40]. Bacterial cultures adjusted to the 0.5 McFarland density standard (1.5 × 10^8^ CFU/mL) were dispensed into the assay plates, reaching a final concentration of 10 µM per compound in a total volume of 200 µL. Assay plates were then incubated at 37 °C for 18–24 h, after which growth inhibition was quantified by measuring optical density (OD) at 595 nm using a Bio-Rad iMark™ microplate reader (Bio-Rad, Hercules, CA, USA). Viability was also assessed with a 1% resazurin solution. Compounds with inhibitory activities greater than or equal to 80% were considered potential antimicrobial compounds. All experiments were performed in duplicate.

### 4.3. Determining the IC and MBC Values of Compounds

The inhibitory concentration (IC) and minimum bactericidal concentration (MBC) values of the Pandemic Response Box (PRB) and COVID Box (CB) compounds were determined using the broth microdilution method [38,41]. In the wells of 96-well plates, two-fold serial dilutions were prepared for the concentration range of 20–0.039 µM. After incubation, each well’s optical density (OD) was measured at 595 nm using a Bio-Rad microplate reader (Bio-Rad, Hercules, CA, USA), and based on the OD values, the inhibition percentages were calculated. MHB served as the negative control. The bacterial isolates were used as positive controls. All experiments were performed in duplicate. Percentage inhibition (%) was calculated using the following equation:$$Inhibition\left(\%\right)=\frac{{OD}_{untreatedcontrol}-{OD}_{treatment}}{{OD}_{untreatedcontrol}}\times 100$$

### 4.4. Survival Curve

The efficacy of each compound at the respective inhibitory concentration was determined over 24 h by measuring absorbance at predetermined intervals followed by plotting the growth curves. A bacterial suspension was adjusted to the 0.5 McFarland density standard (1.5 × 10^8^ CFU/mL) and inoculated into the wells containing the compounds to be evaluated (0, 2, 4, 6, 8, 10, 12, and 24 h). Afterward, the absorbance in each well was measured at 595 nm using an iMark^TM^ Microplate (Bio-Rad, Hercules, CA, USA), and the corresponding bacterial growth curves were plotted. All experiments were performed in duplicate.

### 4.5. Biofilm Formation Inhibition Analysis

The ability of the concerned compounds to inhibit biofilm formation was investigated. Plates were maintained under stable incubation conditions at 37 °C for 24 h to promote bacterial development and biofilm maturation. Subsequently, the planktonic cells were removed, and the biofilms were stained with 0.1% crystal violet [15,16]. Biofilm biomass was then quantified by measuring the optical density at 595 nm using an iMark^TM^ Microplate (Bio-Rad, Hercules, CA, USA) [42]. Percentage inhibition was calculated using the amount of biofilm formed (defined as 100% biofilm) and the sterility control of the medium (defined as 0% biofilm). All experiments were performed in triplicate.

### 4.6. Membrane Integrity Assay: Protein Quantification

The effects of the concerned compounds on bacterial cell membrane integrity were monitored through the quantification of protein leakage. The microplate was incubated at 37 °C for 4 h, following which the contents of each microplate were centrifuged at 2500 rpm for 5 min at 4 °C. The obtained supernatant was then assayed using the Pierce^TM^ BCA Protein Assay Kit (Thermo Scientific, Waltham, MA, USA) followed by OD (595 nm) measurement using an iMark^TM^ Microplate Absorbance Reader (Bio-Rad, Hercules, CA, USA) to determine the protein quantity released from the cytoplasm. All experiments were performed in duplicate.

### 4.7. Statistical Analysis

The results are presented as mean ± standard deviation. The treated and untreated controls were compared using a one-way analysis of variance followed by the Kruskal–Wallis test. A two-tailed *t*-test was conducted to determine the biofilm-associated CFUs in the control and treatment group wells. All statistical analyses and graph plotting were performed using GraphPad Prism (version 8.1). The size for all labels and compound names was chosen to ensure consistency. The chemical structures of compounds were drawn in a standardized manner using the Draw Structure tool available to researchers on PubChem (https://pubchem.ncbi.nlm.nih.gov, accessed on 20 July 2024).

## Figures and Tables

**Figure 1 antibiotics-13-00723-f001:**
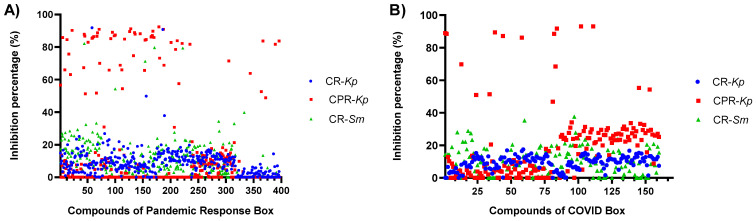
Initial screening results for the boxes. (**A**) Among the PRB compounds evaluated against CPR-*Kp*, 38 exhibited inhibition percentages above 80%, with five exhibiting exceptionally high inhibition rates of above 90%: fusidic acid, MMV1580853, and MMV1634399. Only two PRB compounds exhibited an inhibition percentage above 80%: gepotidacin and MMV1634402 for CR-*Kp*, and against CR-*Sm*, trimethoprim exhibited a remarkable inhibition percentage of 82.14%. (**B**) Among the CB compounds evaluated against CPR-*Kp*, four compounds, oxyclozanide, doxorubicin, simeprevir, and niclosamide, exhibited inhibition rates above 80%.

**Figure 2 antibiotics-13-00723-f002:**
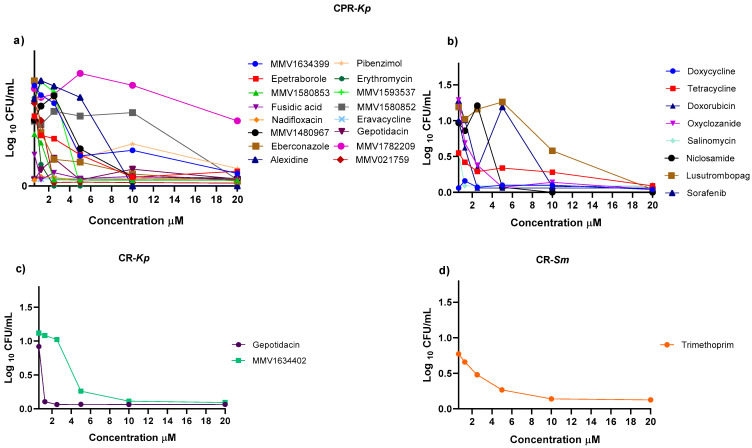
Comparative analysis of the antibacterial activities of PRB and the CB compounds. Dose–response analysis of the growth of *K. pneumoniae* and *S. marcescens*. (**a**) ICs were determined for the compounds from PRB with inhibitory activity against CPR-*Kp*: fusidic acid, MMV1580853, MMV1634399, nadifloxacin, pibenzimol, clofazimine, MMV1593541, and erythromycin. (**b**) ICs were determined for the CB compounds that inhibited the growth of CPR-*Kp*: doxycyline, tetracycline, oxyclozanide, doxorubicin, and niclosamide. (**c**) ICs were determined for the compounds from PRB that inhibited the growth of CR-*Kp*: gepotidacin and MMV1634402. (**d**) ICs were determined for the compound from PRB that inhibited the growth of CR-*Sm*: trimethoprim. All tests were performed in duplicate. Each point represents the mean of duplicate determinations.

**Figure 3 antibiotics-13-00723-f003:**
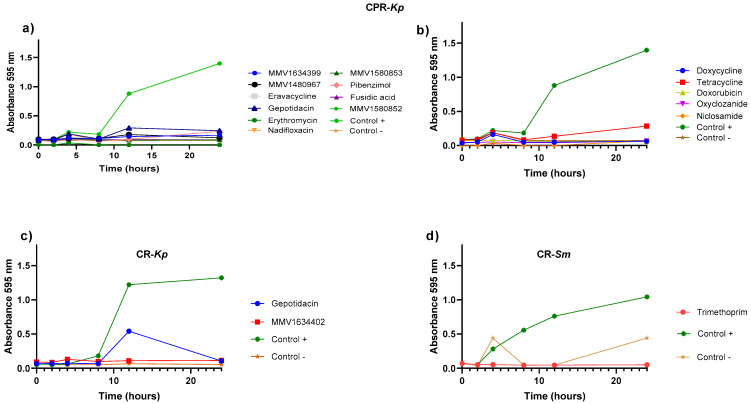
Elimination time kinetics of different antibacterial agents against the growth curves of selected bacterial strains under the effect of different compounds from PRB and CB. (**a**) Survival curve (over 24 h) for CPR-*Kp* when treated with the compounds from PRB: MMV1634399, MMV1580853, fusidic acid, nadifloxacin, MMV1480967, eberconazole, gepotidacin, erythromycin, pibenzimol, eravacycline, and MMV1580852. (**b**) Survival curve (over 24 h) for CPR-*Kp* when treated with the compounds from CB: doxycycline tetracycline, doxorubicin, oxyclozanide, and niclosamide. (**c**) Survival curve (over 24 h) for CR-*Kp* when treated with the PRB compounds: gepotidacin and MMV1634402. (**d**) Survival curve (over 24 h) for CR-*Sm* when treated with the compounds from PRB: trimethoprim. All tests were performed in duplicate. Each point represents the average of duplicate determinations.

**Figure 4 antibiotics-13-00723-f004:**
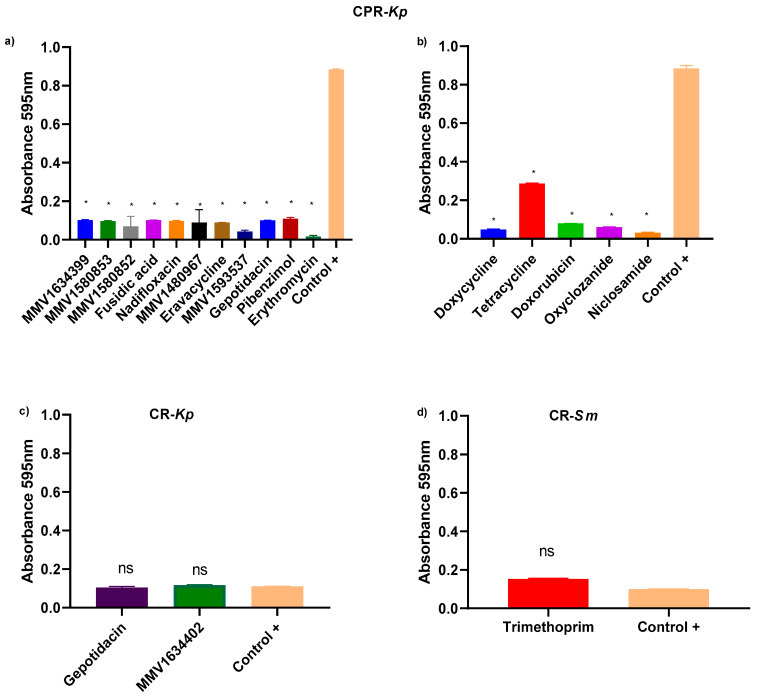
Inhibition of biofilm formation by PRB and CB compounds. (**a**) CPR-*Kp* treated with different PRB compounds exhibiting variable levels of biofilm formation, with only a few compounds demonstrated to significantly reduce biofilm formation, namely, MMV 1634399, MMV 1580853, fusidic acid, nadifloxacin, gepotidacin, erythromycin, pibenzimol MMV 1593537, eravacycline, and MMV1580852. (**b**) The CB compounds doxycycline, tetracycline, doxorubicin, oxyclozanide, and niclosamide exhibited inhibitory effects on biofilm formation in CPR-*Kp* compared to the positive control. (**c**) Gepotidacin and MMV1634402, among the PRB compounds evaluated for biofilm formation against CR-*Kp*, exhibited low efficacy in biofilm inhibition, with values close to those noted for the positive control. (**d**) Trimethoprim from PRB did not exhibit a significant difference in its inhibitory effect on biofilm formation in CR-*Sm* compared to the positive control. All tests were performed in triplicate. Each point represents the average of triplicate determinations. Untreated controls were compared using a one-way analysis of variance (ANOVA) followed by the Kruskal–Wallis test. * *p* value < 0.5; ns: not significant.

**Figure 5 antibiotics-13-00723-f005:**
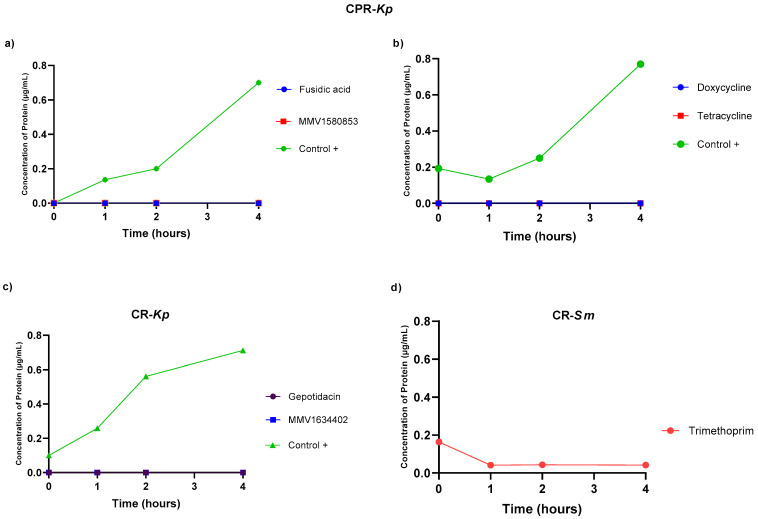
The number of proteins released over time, reflecting the integrity of the cell membrane when the bacterial strains were exposed to the compounds from PRB and CB, are depicted. (**a**) Extravasation of CPR-*Kp* proteins; (**b**) extravasation of CPR-*Kp* proteins; (**c**) extravasation of CR-*Kp* proteins; (**d**) extravasation of CR-*Sm* proteins. The data points represent the averages of the protein concentration values measured at 0, 1, 2, and 4 h post-treatment, providing insights into the temporal dynamics of bacterial membrane damage. All tests were performed in duplicate. Each point represents the average of duplicate determinations.

**Table 1 antibiotics-13-00723-t001:** Antimicrobial susceptibility testing of the strains, showing the resistance profile to multiple antibiotics.

Antibiotics	Minimum Inhibitory Concentration (μg/mL)		
	CPR-*Kp*	CR-*Kp*	CR-*Sm*
Cefepime	>256	Not tested	32
Cefotaxime	>256	Not tested	>256
Ceftazidime	>256	>256	>2
Aztreonam	>32	>32	>32
Imipenem	>16	>16	>16
Meropenem	>16	>16	>8
Ertapenem	>32	>32	>32
Amikacin	64	<8	32
Polymyxin	4	0.5	>64
Tigecycline	<0.5	<0.5	0.5

Antimicrobial susceptibility testing was carried out according to CLSI protocols. MIC: Minimum inhibitory concentration in ug/mL.

**Table 2 antibiotics-13-00723-t002:** Strains, compounds, percentage of inhibition, IC, and chemical structure.

Strains	Compounds	Inhibition (%)	IC (µM)	Disease Set	Chemical Structure
CPR-*Kp*	Fusidic acid	91.28	1.25	Antibacterials	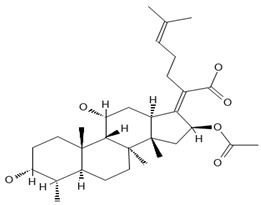
CPR-*Kp*	MMV1580853	90.00	2.5	Antibacterials	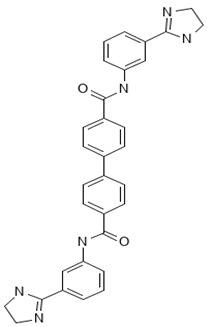
CPR-*Kp*	MMV1634399	90.00	20	Antibacterials	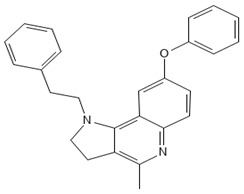
CPR-*Kp*	Nadifloxacin	89.72	0.625	Antibacterials	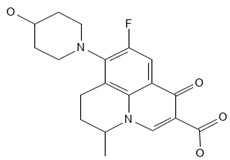
CPR-*Kp*	Pibenzimol	88.75	20	Antibacterials	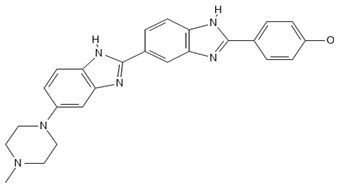
CPR-*Kp*	Erythromycin	89.72	2.5	Antibacterials	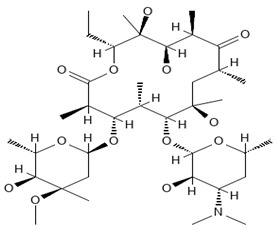
CPR-*Kp*	Doxorubicin	91.74	2.5	Antibacterials	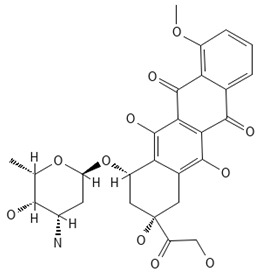
CPR-*Kp*	Niclosamide	89.00	5	Antiparasitic	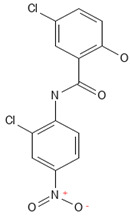
CR-*Kp*	Gepotidacin	91.84	0.625	Antibacterials	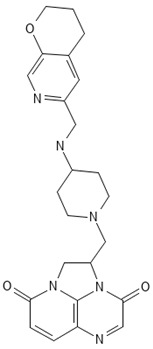
CR-*Kp*	MMV1634402	90.78	10	Antibacterials	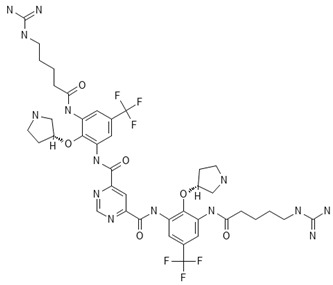
CR-*Sm*	Trimethoprim	82.14	10	Antibacterials	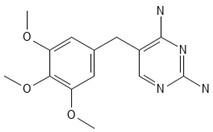

Compounds of the Pandemic Response and COVID Box; Inhibition (%): inhibition percentage; IC: inhibitory concentration.

## Data Availability

The datasets generated during and/or analyzed during the current study are available from the corresponding author on reasonable request.

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
