# Peer review of "Investigating the Antimicrobial Potential of 560 Compounds from the Pandemic Response Box and COVID Box against Resistant Gram-Negative Bacteria"

_antibiotics, 2024, doi:10.3390/antibiotics13080723_

Round 1

Reviewer 1 Report

Comments and Suggestions for Authors

"Investigating the Antimicrobial Potential of 560 compounds from the MMV Pandemic Response Box and COVID Box Against Resistant Gram-Negative Bacteria." si a nice article about the evaluation of 560 compounds against antibiotic resistant strains of K. pneumoniae, P. aeruginosa and S. macescens. The MIC and MBC, inhibition of biofilm formation and membrane integrity assay have been evaluated for the compounds that inhibited bacterial growth of more than 80-90%.

This article is an interesting work of drug repurposing for what I retrieved from the text, not sure if all the identified positive drugs are indeed already approved, so this is something that I suggest to underline. 
Some minor changes may improve the text for example if you put the matherial and methods before the discussion, the acronyms will be explained earlier in the text and it will be easier for the reader.

Nice work. congratulation. 

Author Response

  1. This article is an interesting work of drug repurposing for what I retrieved from the text, not sure if all the identified positive drugs are indeed already approved, so this is something that I suggest underlining. 

Response: Thank you for highlighting this important aspect. In response to the reviewer's comment, we have revised the manuscript to emphasize this point. Specifically, we have rewritten the paragraphs for the two compounds that do not yet have ongoing clinical studies and are not marketed for clinical purposes (Page 8, lines 194 – 209).

“MMV1580853 is a biphenyl bisamidine compound that targets undecaprenyl diphosphate synthase (UPPS), a critical enzyme for the synthesis of the bacterial cell wall. While no clinical studies are currently underway, MMV1580853 disrupts UPPS, hindering the production of essential cell wall components and leading to bacterial cell death. Preclinical evaluations using mouse models of staphylococcal infection demonstrated that MMV1580853 is safe, and shows reproducibility in vivo [21]. These findings reinforce its therapeutic potential against PCR-Kp strains identified in our study.

In the present study, MMV1634399 (4-methyl-8-phenoxy-1-(2-phenylethyl)-2,3-dihydropyrrolo [3,2-c] quinoline), a compound currently without a commercial name or clinical use, has been shown to effectively reduce bacterial growth and inhibit biofilm formation. This suggests that MMV1634399 targets essential bacterial processes. The compound's ability to inhibit biofilms, which protect bacteria from immune system attacks and antibiotics [22], highlights its potential as a valuable candidate for further research. Although MMV1634399 has been little studied previously, it reportedly exhibits inhibitory activity against fungi as well [23]”.

  1. Some minor changes may improve the text for example if you put the matherial and methods before the discussion, the acronyms will be explained earlier in the text and it will be easier for the reader.

Response: We thank the reviewer for the comment and apologize for the confusion. We agree that this adjustment would enhance the clarity of the manuscript. However, the journal's formatting guidelines stipulate that the Results and Discussion sections must precede the Materials and Methods section. To improve readability, we have defined all acronyms from the Abstract through the Discussion sections to assist the reader before they refer to the Materials and Methods section. We have updated the manuscript accordingly to ensure that the acronyms are clearly defined at the earliest mention, enhancing the reader's understanding throughout the text. The acronyms are defined as follows:

“Antimicrobial resistance (AMR)” (Page 1, line 11)

“Multidrug-resistant (MDR) strains” (Page 1, line 15)

"MMV Pandemic Response Box (PRB) and COVID Box (CB)" (Page 1, line 16)

"Minimum inhibitory concentration (MIC)" and "Minimum bactericidal concentration (MBC)" (Page 1, lines 17, 18, and 19)

"Polymyxin-carbapenem-resistant K. pneumoniae (PCR-Kp) strain" (Page 2, line 70)

"Carbapenem-resistant K. pneumoniae (CR-Kp) strain" (Page 2, lines 73 and 74)

"Carbapenem-resistant S. marcescens (CR-Sm) strain" (Page 2, line 86)

Reviewer 2 Report

Comments and Suggestions for Authors

The manuscript by Melo et al. describes testing of several hundreds of compounds available from large Covid-19 and malaria screening campaigns against three gram-negative bacterial strains and their further characterization in MIC, time-kill, biofilm formation and protein leakage assays. The authors identify a few already known antibiotics and other, likely broadly toxic compounds (such as the biphenyl-containing one) as inhibitors of their strains.

Even though potentially interesting observations could be made from these studies for the molecules that are not already well characterized, the experimental rigor of this study is insufficient in using the reported data. There are also major problems with the quality of the presentation, as described below. For these reasons, this manuscript, at present, appears to be premature for publication.

Major comments:

1) There is a concern about scientific rigor: the initial screening was done in duplicate, which is ok. Other testing appears to be done as single experiments, as there are no error bars in any of the plots. This is unacceptable for a publication, and these data cannot be treated as reproducible.

2) Toxicity to mammalian cells should be tested as a counterscreen for the compounds that are not known antibiotics.

3) Protein leakage is a sign of a very severe rupture of the cell envelope. The absence of protein leakage does not mean that cell envelope is intact. This needs to be explained.

4) There is extensive literature on the known antibiotics (such as fusidic acid), which is not given enough attention in the Discussion.

5) The manuscript is not well organized making it difficult to read. In the first section of Results, the authors need to 1) provide a table of antibiotic susceptibilities for each strain tested (since these are not commercially available strains), 2) mention that the inhibition was measured at 10 microM of compound and 3) provide a table containing a chemical structure, %inhibition and MIC value for each strain for the compounds described in this section. The MIC values then need to be removed from figure legend for Figure 2, because they do not belong there.

6) Figures need to be made at high resolution. The figures also need to be carefully and uniformly formatted, which is currently not the case. Also, please scale up the figures to minimize empty spaces; this will also make them easier to view.

Minor comments

-Specify the source of the bacterial strains tested.

-Need to describe briefly the compounds that were tested in terms of chemical diversity, known biological activities etc, even if they are from known libraries.

-Expression: reducing biofilm density is inaccurate, because the compounds were not tested against preformed biofilm. The description needs to be worded in terms of inhibition of biofilm formation.

Line 94: clarify what is meant by “each compound”

Abstract, line 15, please use the word potency instead of efficacy here and throughout, when referring to activity of molecules against bacteria in vitro.

Line 23: potency, instead of potential.

Line 61: use full name for MMV on the first use.

Line 65: delete the format guidelines left over from the draft.

Comments on the Quality of English Language

Some minor English errors need to be corrected.

Author Response

  1. There is a concern about scientific rigor: the initial screening was done in duplicate, which is ok. Other testing appears to be done as single experiments, as there are no error bars in any of the plots. This is unacceptable for a publication, and these data cannot be treated as reproducible.

Response:  We appreciate the reviewers' comments. Due to the limited availability of compounds from the MMV Pandemic Response Box and COVID Box, coupled with the number of bacterial strains and tests conducted, both our initial screening and subsequent assays were performed in duplicate. The screening methodology followed that described by Sivasankar et al., (2023), which did not incorporate controls at the initial stage. The primary aim of this screening was to assess the antimicrobial activity across the 560 compounds. Subsequently, compounds demonstrating the most promising antimicrobial activity were selected for further testing, where appropriate controls were rigorously implemented. To ensure transparency and uphold the validity of our findings, we have incorporated this information into the manuscript. Also, in response to the reviewer’s comment, we have included the following statements in the discussions to explicitly state this limitation in our study:

            “Although our results were promising, this study had some limitations, including a limited number of strains tested and the assays performed in duplicate.”

  1. Toxicity to mammalian cells should be tested as a counterscreen for the compounds that are not known antibiotics.

Response:  Thank you for your feedback. The primary purpose of the MMV Pandemic Response Box and COVID Box is drug repositioning, similar to Phase I trials, since these drugs are already intended for other purposes. The two compounds, MMV1580853 and MMV1634399, which are still understudied showed antimicrobial activity. In our literature review, we discovered that MMV1580853 has a published preclinical study in rodents, demonstrating safety in this model. This indicates that MMV1580853 is a promising candidate for further studies to provide a comprehensive view of its activity, mechanism, dosage, and safety in humans. The aim of our study was an initial screening. Future studies will incorporate mammalian cell toxicity tests to ensure a thorough assessment of these compounds' safety profiles.

  1. Protein leakage is a sign of a very severe rupture of the cell envelope. The absence of protein leakage does not mean that cell envelope is intact. This needs to be explained. on the

Response:  The suggestion was accepted. In response to the reviewer’s comment, this information was added in the discussion section that provides detailed information about the membrane integrity test and the possible interpretations of the results (lines 179-182).

“To identify possible mechanisms of action of the compounds tested, we conducted the protein extravasation test, which is often used as an indicator of the integrity of the cell membrane. This leakage means significant damage to the cell envelope, which includes the cell membrane and, in some cells, the cell wall. Minor damage may not lead to detectable protein leakage but can still impair cell function and viability. Furthermore, cells have mechanisms to quickly repair minor damage to the envelope, preventing protein leakage. However, the cell envelope can still allow leakage of smaller molecules, such as ions and metabolites, without proteins. Therefore, the absence of protein extravasation in our assays does not rule out the presence of damage [21,22].”

  1. The manuscript is not well organized making it difficult to read. In the first section of Results, the authors need to 1) provide a table of antibiotic susceptibilities for each strain tested (since these are not commercially available strains), 2) mention that the inhibition was measured at 10 microM of compound and 3) provide a table containing a chemical structure, %inhibition and MIC value for each strain for the compounds described in this section. The MIC values then need to be removed from figure legend for Figure 2, because they do not belong there.

Response: Thank you for your feedback on the manuscript. We have added a table 1 of antibiotic susceptibilities for each strain tested. Additionally, we have included a table containing the chemical structure, percentage of inhibition, and Minimum Inhibitory Concentration (MIC) value for each strain for the compounds described in this section. We also specified that inhibition was measured at a concentration of 10 µM of the compound. Furthermore, the legend in Figure 2 has been revised to remove the MIC values, as they do not belong there. We believe these changes will improve the organization and readability of the manuscript. Additionally, data from P. aeruginosa was excluded from this document. This exclusion was aimed at narrowing the study's focus exclusively to bacteria from the Enterobacteriaceae family. Data pertaining to P. aeruginosa will be further refined and could be considered for future publications.

  1. Figures need to be made at high resolution. The figures also need to be carefully and uniformly formatted, which is currently not the case. Also, please scale up the figures to minimize empty spaces; this will also make them easier to view.

Response: We apologize for the confusion and have revised the figures. We recognize that the legibility of names in figures is essential for the presentation of our data. Therefore, we have increased the font size of all text in the figures. Additionally, we have improved the resolution and adjusted the layout to avoid overcrowding, ensuring all elements are clearly visible. We appreciate your valuable suggestion to improve the quality of the figures in the manuscript.

MINOR COMMENTS

  1. Figures need to be made at high resolution. The figures also need to be carefully and uniformly formatted, which is currently not the case. Also, please scale up the figures to minimize empty spaces; this will also make them easier to view. Specify the origin of the bacterial strains tested.

Response: The suggestion was accepted, and information was added in the Materials and Methods Section (Page 12, lines 283-284).

  1. It is necessary to briefly describe the compounds that were tested in terms of chemical diversity, known biological activities, etc., even if they are from known libraries.

Response:  The suggestion was accepted, and information was added (Page 13, lines 297-298).

  1. Expression: reducing biofilm density is inaccurate as the compounds were not tested against preformed biofilm. The description needs to be written in terms of inhibition of biofilm formation.

Response:  The suggestion was accepted, and the changes were performed (Page 1, line 21; Page 3, lines 103-110; Figure 4, page 9, lines 159-169).

  1. Line 94: clarify what is meant by “each compound”.

Response: The suggestion was accepted.

  1. Abstract, line 15, please use the word potency instead of efficacy here and elsewhere when referring to the activity of molecules against bacteria in vitro.

Response: The suggestion was accepted.

  1. Line 23: power rather than potential.

Response: The suggestion was accepted.

  1. Line 61: use full name for MMV on first use.

Response: The suggestion was accepted.

  1. Line 65: Delete the format guidelines left over from the draft.

Response: The suggestion was accepted.

Thank you for the detailed and thoughtful correction.

Reviewer 3 Report

Comments and Suggestions for Authors

I rate the reviewed article well, although it requires a few corrections. The authors tested 560 compounds against 3 species of multidrug-resistant bacteria: Klebsiella pneumoniae, Pseudomonas aeruginosa, and Serratia marcescens. They used the correct methodology, compliant with CLSI, MIC testing, MBC and anti-biofilm activity with crystal violet.

I have 4 critical comments:

1. although 560 potentially antibacterial compounds were tested, I lacked a positive control, i.e. a 100% antibacterial substance, e.g. octenidine, to compare whether the activity is better or worse than the positive control,

2. why are MIC values ​​reported in μM? In microbiology, according to CLSI and EUCAST, MICs are given in mg/L or μg/mL !!! Therefore, MIC and MBC values ​​should be changed and given in microbiological units μg/mL, rather than chemical units μM. This will be more readable and comparable to most antibiotic articles that report mg/L or μg/mL

3. the figures are too small and illegible, e.g. in Fig. 2 drug names are so small that they will probably be invisible or illegible in the final version of the article

4. bacterial names should be written in italics

Comments on the Quality of English Language

Good

Author Response

  1. Although 560 potentially antibacterial compounds were tested, I lacked a positive control, i.e. a 100% antibacterial substance, e.g. clonidine, to compare whether the activity is better or worse than the positive control.

Response:  Thank you for your insightful comment. The primary focus of this phase was a preliminary screening of 560 compounds from the MMV Pandemic Response Box and COVID Box to identify potential antibacterial activity against multidrug-resistant strains of Klebsiella pneumoniae, and Serratia marcescens. The inclusion of a positive control, such as clonidine, was not prioritized at this stage, as our objective was to identify promising candidates for more detailed studies later.  The screening assay was performed by described for Sivasankar et al., 2023, which does not use a known bactericidal compound, and uses bacterial growth inhibition cuts of 80%, and comparison between the inhibition potential between them. After screening, we used positive and negative controls in all tests performed, except for the membrane extravasation test for the resistant strain of S. marcescens where no drug showed activity on the bacterial membrane. This feedback will contribute to refining our experimental procedures in future phases of research, ensuring the inclusion of appropriate controls in future protocols.

  1. Why are MIC values ​​reported in μM? In microbiology, according to CLSI and EUCAST, MICs are given in mg/L or μg/mL !!! Therefore, MIC and MBC values ​​should be changed and given in microbiological units μg/mL, rather than chemical units μM. This will be more readable and comparable to most antibiotic articles that report mg/L or μg/mL.

Response: We appreciate the reviewers' constructive comments to substantially improve the readability of our manuscript. We apologize for the confusion and recognize the importance of adhering to established microbiological standards for better comparability with other studies. Thus, we converted the MIC and MBC values from μM to μg/mL, following the formula:

Concentration (μg/mL) = µM = (µg/mL / MW) * 1000

Thank you for helping us improve the quality and presentation of our research. We have updated the text in the manuscript accordingly.

  1. The figures are too small and illegible, e.g. in Fig. 2 drug names are so small that they will probably be invisible or illegible in the final version of the article.

Response: We apologize for the confusion and have revised the figures. We have increased the font size of all text in the figures. Additionally, we have improved the resolution and adjusted the layout to avoid overcrowding, ensuring all elements are clearly visible. We appreciate your valuable suggestion to improve the quality of the figures in the manuscript.

  1. Bacterial names should be written in italics.

Response: We apologize for the confusion and have made adjustments throughout the manuscript to comply with the standard formatting of scientific names by writing the names of bacteria in italics. We appreciate your attention to detail.

Round 2

Reviewer 2 Report

Comments and Suggestions for Authors

Major comments:

1) A serious issue that emerged from the new Table 2 is with the MIC, which is conventionally defined as a minimum concentration of the compound at which no growth can be detected. The MIC values in the Tables are clearly wrong for that definition, if one compares them with Fig 2 c and d. Also, there should only be one MIC value. There is no such thing as “MIC of inhibition”. This value needs to be clearly defined separately, if the authors still wish to show it.

2) The comment about the insufficient replicates was not adequately addressed. Both replicate data points need to be shown in the figures, instead of a single point, for all experiments, either in the main text or the supplementary information. This is the way to ethically show duplicates.

Minor comments:

3) The comment about toxicity to the mammalian cells is not adequately addressed. Even if the drugs are repurposed, the dose relevant for the antibiotic activity may be higher than for other indications, where the molecules can be toxic, so the cytotoxicity tests are still needed. This should be at least discussed.

4) Figures are still not properly formatted. For example, Figure 2: the x-axis is not correctly labeled in all panels. This should be compound concentration, specify units. The panel size is still not the same for all panels, lines have different thicknesses everywhere. Figure titles should be moved to insets (panel b does not have a title). Font size is different for compound names and axis labels in all panels. The authors need to pay more attention to details.

5) Table 2 need to be reformatted carefully. The chemical structures all need to be made in ChemDraw and formatted uniformly (!) in the ASC format, which is common to most journals.

Author Response

  1. A serious issue that emerged from the new Table 2 is with the MIC, which is conventionally defined as a minimum concentration of the compound at which no growth can be detected. The MIC values in the Tables are clearly wrong for that definition, if one compares them with Fig 2 c and d. Also, there should only be one MIC value. There is no such thing as “MIC of inhibition”. This value needs to be clearly defined separately, if the authors still wish to show it.

    Response:  

    Thank you for highlighting this important aspect. We have reformulated Table 2 by incorporating the Inhibitory Concentration (IC) values, obtained through survival curve analysis. The efficacy of each compound at the respective inhibitory concentration was determined over 24 hours by measuring absorbance at predetermined intervals, followed by plotting the growth curves. We have removed the calculation of concentration in µg/mL and adopted solely the unit utilized during the tests. We therefore based our approach on the work of Kim et al., 2021, who used the same percentage of inhibition (80%) and reported IC50 and IC90 in their study. Additionally, we have included the initial function of each compounds. The chemical structures of each compound have been redrawn in a standardized manner using the Draw Structure tool available to researchers on PubChem (https://pubchem.ncbi.nlm.nih.gov). Some structures require more space for better visualization, and therefore, the size of each structure may vary to enhance their clarity (Pages 5-8, lines 136-137).

  2. The comment about the insufficient replicates was not adequately addressed. Both replicate data points need to be shown in the figures, instead of a single point, for all experiments, either in the main text or the supplementary information. This is the way to ethically show duplicates.Response: We appreciate your valuable comment regarding the insufficient replicates in our experimental data. We added the number of repetitions performed for each test in each section of the Materials and Methods (Page 14, lines 321,320; Page 15, lines 342,352,361). Additionally, we include this information in the legends of all graphs (Page 5, line 134; Page 9, line 148; Page 11, lines 160-161). This addition ensures that our methodology is reported transparently and ethically. Thanks again for your feedback.
  3. The comment about toxicity to the mammalian cells is not adequately addressed. Even if the drugs are repurposed, the dose relevant for the antibiotic activity may be higher than for other indications, where the molecules can be toxic, so the cytotoxicity tests are still needed. This should be at least discussed.

    Thank you for your valuable comment regarding toxicity to mammalian cells. In response, we have revised the paragraph to emphasize that these results represent a very early phase in the comprehensive process necessary to commercialize new medicines. Even if medications are reused, it is essential to consider that the dose required for antibiotic activity may be higher than that used for other indications, potentially resulting in toxicity. Therefore, cytotoxicity tests are essential and should be discussed.

    Revised paragraph: “The present study demonstrated that MMV1634399 is an effective agent for the reduction of bacterial growth and the inhibition of biofilm formation. This indicates that the compound targets essential bacterial processes that are independent of any impact on the cell membrane. The capacity of MMV1634399 to repress biofilm formation, which serves to protect bacteria against assaults from the immune system and antibiotics, underscores its potential as a valuable contender for future investigation. Drug repurposing, or reprofiling, is the process of identifying new therapeutic applications for existing pharmaceutical agents. While in vitro (test tube or cell culture) and pre-clinical (animal) studies can provide promising initial results, the process of translating these findings into clinical use is inherently rigorous and lengthy. This process comprises several phases of clinical trials, the objective of which is to guarantee the drug's therapeutic efficacy and safety for human use [28–30]. The drug discovery and development are considered complex, time-consuming, and resource-intensive. It requires multidisciplinary expertise and innovative approaches [31].” (Page 13, line 235-248).

    1. Figures are still not properly formatted. For example, Figure 2: the x-axis is not correctly labeled in all panels. This should be compound concentration, specify units. The panel size is still not the same for all panels, lines have different thicknesses everywhere. Figure titles should be moved to insets (panel b does not have a title). Font size is different for compound names and axis labels in all panels. The authors need to pay more attention to details.
  4.  Table 2 need to be reformatted carefully. The chemical structures all need to be made in ChemDraw and formatted uniformly (!) in the ASC format, which is common to most journals.

    We appreciate your evaluation regarding the formatting of the figures and tables.

    Regarding Figure 2: We have updated this to indicate the compound concentration and specify the units correctly.  Panel sizes have been adjusted to have the same line thickness on all charts. Figure titles have been adjusted. The font size for compound names has been standardized as has the font size for all labels and compound names to ensure consistency. (Page 4, lines 112 and 124; Page 9, line 141; Page 10, line 150, Page 11, line 162)

    For Table 2, we will perform the following revisions:

    The chemical structures of each compound have been redrawn in a standardized manner using the Draw Structure tool available to researchers on PubChem (https://pubchem.ncbi.nlm.nih.gov). Some structures require more space for better visualization, and therefore, the size of each structure may vary to enhance their clarity (Pages 5-8, lines 136-137).

Reviewer 3 Report

Comments and Suggestions for Authors

The authors significantly corrected the manuscript according to the reviewer's suggestions. Recently, I recommend the article for publication.

Author Response

Comments: The authors significantly corrected the manuscript according to the reviewer's suggestions. Recently, I recommend the article for publication.

Response: Thank you for your thorough review and constructive feedback on our manuscript. We diligently addressed all suggestions and made significant corrections as recommended. We appreciate your time and effort to improve the quality of our work.